# Periodic Density Functional Theory (PDFT) Simulating Crystal Structures with Microporous CHA Framework: An Accuracy and Efficiency Study

Xiao-Fang Chen

Institute of Molecular Sciences and Engineering, Institute of Frontier and Interdisciplinary Science, Shandong University, Qingdao 266237, China; chen_smiling@163.com

**Abstract:** The structure property is the fundamental factor in determining the stability, adsorption, catalytic performance, and selectivity of microporous materials. Seven density functional approximations (DFAs) are used to simulate the crystal structure of microporous material for examining the efficiency and accuracy. In comparison with the existing zeolites, microporous materials with CHA framework are selected as the testing model. The calculation results indicate that the least lattice volume deviation is $5.18/2.72$ Å$^3$ from PBE_mGGA, and the second least is $-5.55/-10.36$ Å$^3$ from LDA_PP. Contrary to USPP_LDA, PBE_GW, PAW_PBE, and PAW_GGA overestimate the lattice volume by ~15.00–20.00 Å$^3$. For each method, RMS deviations are less than 0.016 Å for bond length and less than 2.813° for bond angle. To complete the crystal structure calculation, the CPU time reduces in order of USPP_GGA > PBE_GW > PAW_GGA, PBE_mGGA > PAW_PBE > LDA_PP > USPP_LDA. For two testing models, when the calculation time is not important, PBE_mGGA is the best choice, and when the tradeoff between accuracy and efficiency is considered, LDA_PP is preferred. It seems feasible and efficient to simulate the zeolite structure through E-V curve fitting, full optimization, and phonon analysis by the periodic density functional theory.

**Keywords:** structure; microporous material; pseudopotential; Birch–Murnaghan equation; density functional theory; accuracy; calculation duration

## 1. Introduction

Microporous materials have received rising attention in the fields of industry and academia, known for their good stability, excellent adsorption, highly catalytic performance, and the unique selectivity [1–9]. They are usually used as the ion change material, the adsorbent, the sequestering agent, and the catalyst. In addition to the discovery of natural species, the microporous materials are changing with the fast development of artificial synthesis. A series of new species have been synthesized with the new framework or the diverse composition [10–14]. Most of them contain the complex channel and the elusory cavies [15] with a specific three-dimensional topology. Those unique structural features directly determine the physicochemical properties of materials [16–18]. The neutron scattering diffraction (NSD) and the X-ray diffraction of a single crystal or powder sample could provide useful information, such as atomic coordinates and occupation probabilities in the unit cell. The electron microscopy is capable of observing the fine internal structure, such as the faults, weak superlattice, or confine phenomenon. The diffraction microscopy and the electron microscopy have been widely employed to identify the microporous material structure [19–22]. Under the hydrothermal synthesis condition, the high reactivity makes the monitor of the crystal nucleus formation of microporous materials become difficult [23]. The crystallization of microporous materials usually demands harsh experimental conditions with tremendous efforts [24].

The data from the diffraction experiment could be used for structural determination in theory. Among the diverse theoretical methods, the periodic density functional theory

(PDFT) is a prevalent method to simulate the electronic structure of bulks [25–30], although some nonperiodic methods (such as DFT, hybrid DFT:MP2, and QM/MM) have been used to study the adsorption and the chemical reaction of some cluster model zeolites [31–34]. The geometry of materials is determined by the electronic ground-state energy and its variation around the nuclear position [35]. The computational cost increases exponentially with the number of electrons (N), and the calculation of materials with large sizes almost becomes impractical [35,36]. First-principles DFT determines the electronic structure by introducing the electronic density [36]. This method uses the plane-wave basis set and the pseudopotential approximation to resolve the Kohn–Sham equation [37]. Several density functional approximations (DFAs) are adopted to simplify the wave function by the relatively flexible pseudo potential rather than the genuine potential. The concept of pseudopotential was originally put forward by H. Hellmann [38,39]. The introduction of pseudopotential largely reduces the calculation difficulty of the Kohn–Sham equation and saves the computing time. Many pieces of literature documented the history, development, success, and limitation of the pseudopotential [36,40–45].

The pseudopotentials available in VASP packages could be classified into (i) ultrasoft pseudopotential (USPP) and (ii) projector augmented wave (PAW) according to the method. Compared with PAW, USPP is with a higher energy cut-off and is more flexible (or soft) yet has lower precision. Above LDA and GGA, the higher level of pseudopotential called meta-GGA (also abbreviated as mGGA) appears as the third rung of Jacob's ladder, in which the kinetic energy is also considered during the energy density calculation. The meta-GGA method demands tremendous calculation resources, leading to the highest precision. On the other hand, in terms of the exchange-correlation functional, the pseudopotentials could be classified into (i) the simplest approximation and the "mother" of almost all approximations in DFT—local density approximation (LDA), in which the exchange-correlation energy density is regarded as local, and (ii) a generalized gradient approximation (GGA) consisting of two styles—PBE and PW91, where the density at a given position is considered to change gradually [36]. The combination of those two pseudopotential classifications would lead to several formats of density functional approximations, such as (a) PBE-mGGA, (b) PBE_GW, (c) PBE_GW, (d) PAW_GGA, (e) LDA_PP, (f) USPP_GGA, and (g) USPP_LDA. For detailed information about those abbreviations, please refer to the section of Section 3.

In general, with the higher order approximation, the calculation results are more accurate, and the computation cost is higher. However, the calculation time is precious, and the computation resource available is usually limited. The accuracy and efficiency are two factors of high interest during the numerical simulation [46]. In this study, I attempt to obtain some practical information on calculation accuracy and computation cost on DFAs simulating the crystal structure of microporous materials. All-silica chabazite $(Si_{12}O_{24})$ and $AlPO_4$-34 framework [47] are selected as the testing model zeolite. The main reasons for this selection is due to the fact that (1) they are the framework of the existing zeolites of chabaziteHSSZ-13 and HSAPO34 [48]; (2) rich experiment data on HSSZ-13 and HSAPO34 were documented in the literature; (3) HSSZ-13 and HSAPO34 have wide applications in industry, such as adsorption [49,50], toxic molecule capture [50–52], methanol-to-olefin [2,53,54], syngas-to-light olefins [55,56] phase transformation to/from other structures (for examples: DNL-6 [57], FAU [58], and SAPO-5 [59]), and so on; (4) two models contain the vast majority of elements of zeolites, such as Al, Si, O, and P; (5) Bronsted acid proton is very small in size and has little influence on the zeolitic topology [60]. This study would be helpful for the researcher to use DFA to predict zeolitic crystal structure by VASP packages. The screening method seems routine but effective in obtaining detailed information on the precision and the cost of DFAs on the model zeolite.

## 2. Results and Discussion

### 2.1. Topology

All-silica chabazite (Figure 1a) and $AlPO_4$-34 framework (Figure 1c) are with the CHA topology. The single unit cell (1UC) of the CHA framework with the rhombohedral

representation contains 36 atoms, including 12 tetrahedral atoms and 24 oxygen atoms. The chemical formula is $Si_{12}O_{24}$ and $Al_6P_6O_{24}$ in 1UC, respectively. The Al and P positions alternately appear, then the —(O-Al-O-P)$_n$— chemical chains are formed in the $AlPO_4$-34 framework. The space group reduces from R3M in all silica chabazite to R3 in $AlPO_4$-34. Whether the display range extends along three crystallographic directions, A, B, and C, the channel and the cavity would exhibit it (Figure 1b). There is an 8-member ring perpendicular to the xz or yz crystalline plane and a 12-member ring along the side face of the 8-member ring. The oxygen atom in either all-silica chabazite or $AlPO_4$-34 is classified into O(1), O(2), O(3), and O(4). O(1) is shared by one 8-O ring and two 4-O rings; O(2) is shared by three rings of one 8-O, one 6-O, and one 4-O; O(3) is shared by one 6-O ring and two 4-O rings; and finally, O(4) is owned to two 8-O rings and one 4-O ring. It is worth noting that the hierarchical structure of real zeolite SAPO-34 is influenced by differential Si distribution and desilication [61].

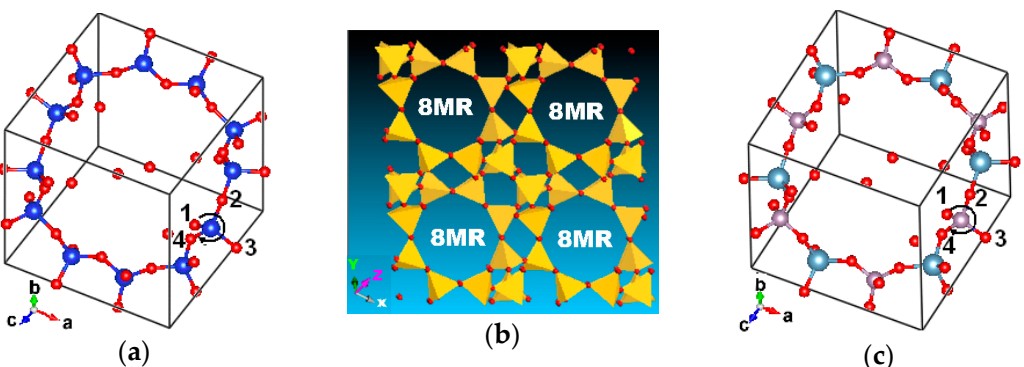

**(a)**                               **(b)**                               **(c)**

**Figure 1.** Structures with CHA framework: (**a**) chabazite (**b**) CHA (**c**) $AlPO_4$-34. Notes: O in red, N in blue, Al in cyan, and P in purple.

*2.2. All-Silica Chabazite*

2.2.1. Energy-Volume Curve and Fitted Equilibrium Volume ($V_{0,fitted}$)

Figure 2 lists the dependence of the total energy on the unit-cell volume for all-silica chabazite under seven sets of pseudopotentials. As shown, seven curves all exhibit the parabola shape, and the pseudopotential category has an obvious influence on the trend of the E-V curve. The total energy of each curve at 750 $Å^3$ is lower than that at 900 $Å^3$.

$$E(v) = E_0 + \frac{9V_0B_0}{16}\left\{ \left( \left(\frac{V_0}{V}\right)^{\frac{2}{3}} - 1 \right)^3 B_0' + \left( \left(\frac{V_0}{V}\right)^{\frac{2}{3}} - 1 \right)^2 \left( 6 - 4\left(\frac{V_0}{V}\right)^{\frac{2}{3}} \right) \right\} \quad (1)$$

The equilibrium volume ($V_0$) and the equilibrium energy ($E_0$) under each set of pseudopotentials can be obtained by fitting the E-V curve with the third order Birch–Murnaghan equation of state (Equation (1)). Herein, E(V) is the total energy at the constant volume (V), $E_0$ is the equilibrium electronic energy, and $V_0$ is the equilibrium unit-cell volume. $B_0$ and $B_0'$ are the bulk modulus and its derivative with respect to pressure, respectively.

The fitted unit-cell volume ($V_{0,fitted}$) is listed in Table 1. Obviously, there is divergence between them. A powder neutron diffraction study indicates the single unit-cell volume of chabazite HSSZ-13 is about 792.32 $Å^3$ [62]. By comparison, PBE_mGGA gives rise to $V_{0,fitted}$ most close to this experimental value, followed by LDA_PP. The fitted $V_0$ from PAW_GGA [48], PAW_PBE, and PBE_GW are about 19.00 $Å^3$ above the experimental value. The biggest deviation of fitted $V_0$ is from USPP_LDA.

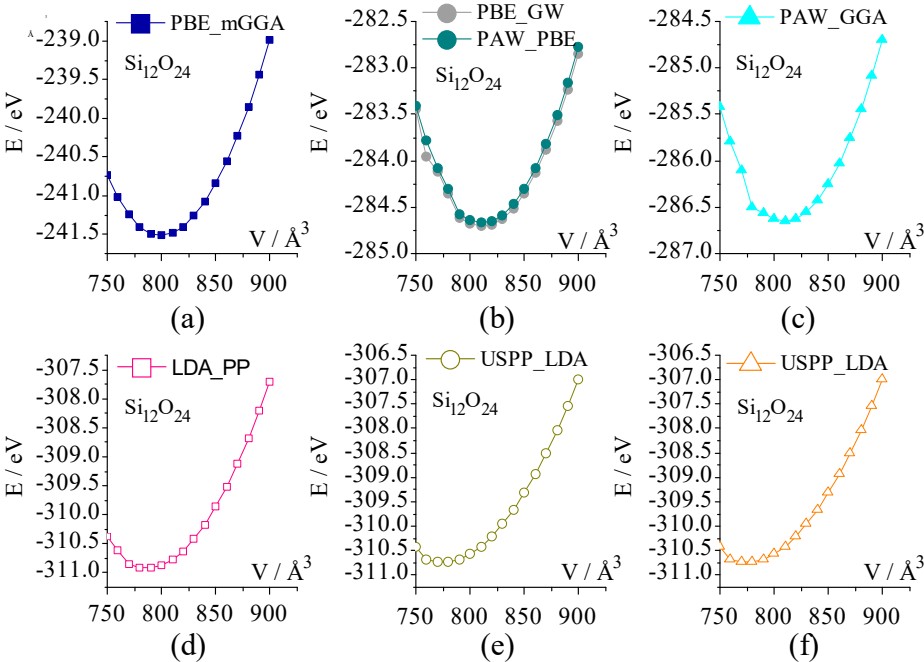

**Figure 2.** The total energy dependence on the unit-cell volume of all-silica chabazite calculated by seven sets of pseudopotentials: (**a**) PBE_mGGA, (**b**) PBE_GW and PAW_PBE, (**c**) PAW_GGA, (**d**) LDA_PP, (**e**) USPP_GGA, and (**f**) USPP_LDA.

**Table 1.** Lattice parameters of all-silica chabazite.

| Pseudopotential | $a$/Å | $\alpha$/° | $V_0$/Å³ | $V_{0,fitted}$/Å³ |
|---|---|---|---|---|
| PBE_mGGA | 9.31 | 94.59 | 797.61 | 799.75 |
| PBE_GW | 9.35 | 94.28 | 810.89 | 810.90 |
| PAW_PBE | 9.35 | 94.40 | 810.64 | 811.84 |
| PAW_GGA | 9.35 | 94.34 | 809.12 | 810. 80 |
| LDA_PP | 9.26 | 94.40 | 786.88 | 789.13 |
| USPP_GGA | 9.31 | 94.21 | 800.36 | 801.54 |
| USPP_LDA | 9.21 | 94.18 | 775.43 | 777.39 |
| Exp [62] | 9.28 | 94.27 | 792.32 | - |

2.2.2. Precise Structure of All-Silica Chabazite and Calculation Error

The full optimization is carried out to obtain a more precise equilibrium structure. The initial structure selected is with the lattice volume near to $V_{0,fitted}$ in Table 1. The precise lattice parameters of all-silica chabazite are also listed in Table 1. There is a = b = c and $\alpha = \beta = \gamma$ for space group of R3m.

In comparison with zeolite chabazite HSSZ-13 [62], PBE_mGGA gives rise to the lattice volume closest to the experimental volume, with an increment of 5.18 Å³; the volume decrement is only 5.55 Å³ for LDA_PP; but PAW_PBE, PBE_GW, and PAW_GGA result in the bigger positive volume deviation ranging from 16.69–18.46 Å³; two ultrasoft types of pseudopotentials lead to a volume deviation of +9.11 Å³ and −15.05 Å³, respectively. In the case of lattice edge (a), PBE_mGGA produces an increment of 0.03 Å, same as USPP_GGA; the least deviation is −0.02 Å from LDA_PP; PAW_PBE, PBE_GW, and PAW_GGA all overestimate the lattice edge by 0.07 Å, contrary to USPP_LDA. Overall, the lattice angle deviations are not big, with the absolute value below 0.32°.

The bond length and the bond angle in crystalline could reflect the ion position and the bonding format. The previous neutron scattering experiment [62] showed that, for zeolite chabazite HSSZ-13, the Si-O bond length is 1.600–1.617 Å, ∠T-O-Si ranges from 144.73–149.94°, and ∠O-T-O is in the range of 107.88°–110.33° (Table S1 in the Supplementary Material). Figure 3 lists the calculation deviation of bond length or bond angle from

the experiment value [62]. At a glance, the dot zigzag shapes are similar to each other whether the *X*-axis is omitted.

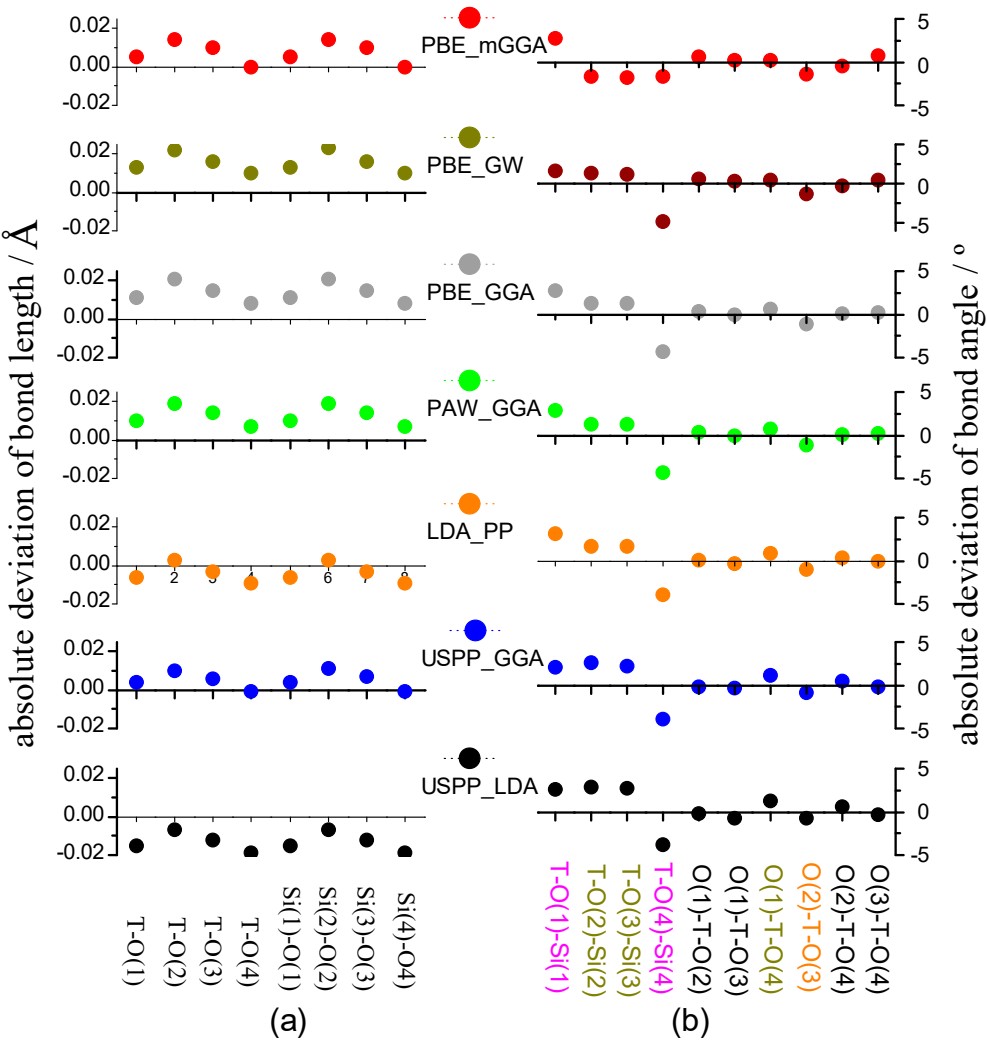

**Figure 3.** The calculation deviation of (**a**) bond length and (**b**) bond angle for all-silica chabazite (T = Si).

Figure 3a shows that PBE_mGGA, PBE_GW, PAW_PBE, and PAW_GGA slightly overestimate all Si-O bond lengths; LDA_PP weakly underestimates all bond lengths of Si-O, except Si-O(2); USPP_GGA overestimates all bond lengths of Si-O, except Si-O(4); and USPP_LDA underestimates all Si-O bond lengths. The absolute values of bond length deviation are as follows: PBE_mGGA, PBE_GW, PAW_PBE, and PAW_GGA, ≤0.020 Å; LDA_PP, ≤0.009 Å; USPP_GGA, ≤0.011 Å; and USPP_LDA, 0.019 Å, respectively.

As far as the bond angle deviation is concerned, as shown in Figure 3b, ∠T-O-Si is generally bigger than ∠O-T-O. Two angles (magenta) have the bigger deviation, whose absolute values are 1.662°–3.190° for ∠T-O(1)-Si(1) and 3.748°–4.872° for ∠T-O(4)-Si(4), respectively. In order of USPP_LDA, USPP_GGA, LDA_PP, PAW_GGA, PAW_PBE, and PBE_GW, the deviation of three angles (dark yellow) of ∠T-O(2)-Si(2), ∠T-O(3)-Si(3), and ∠O(1)-T-O(4) decreases while ∠O(2)-T-O(3) (orange) increases. Four bond angles (black) are little influenced by the PDFT methods selected, including ∠O(1)-T-O(2), ∠O(1)-T-O(3), ∠O(2)-T-O(4), and ∠O(3)-T-O(4).

PBE_mGGA gives rise to small bond angle deviations, whose absolute value is less than 1.421° for ∠O-T-O and ranges from 1.604°–2.673° for ∠T-O-Si. The absolute values of bond angle deviations of ∠O-T-O/∠T-O-Si are as follows: PAW_PBE,

$1.235° – 4.399° / 0.004° – 1.195°$; and LDA_PP: $1.14° – 3.190° / 0.041° – 1.030°$. In total, the absolute value of bond angle deviation of all-silica chabazite is within $4.872°$.

The root mean square (RMS) deviation is further calculated to assess the calculation error. Table 2 shows thatall-silica chabazite has a narrow range of RMS deviation of either bond length or bond angle, one is $0.006 – 0.016$ Å, and the other is $1.382° – 1.990°$. This implies that, the theoretical bond length or bond angle at each of pseudopotential is close to the corresponding experimental value. For all-silica chabazite, among seven DFAs, PBE_mGGA produces the least amount of RMS deviation (i.e., $1.382°$)of bond angle, and LDA_PP brings about the least RMS deviation (i.e., $0.006$Å) ofbond length. The RMS deviations from PAW_PBE are in the middle of those from all pseudopotentials.

**Table 2.** The root mean square (RMS) deviation of bond length ($R_{T-O}$) and bond angle (A) for the all-silica chabazite.

| Pseudopotential | Chabazite | | AlPO$_4$-34 | |
|---|---|---|---|---|
| | $R_{T-O}$/Å | A/° | $R_{T-O}$/Å | A/° |
| PBE_mGGA | 0.016 | 1.382 | 0.066 | 2.310 |
| PBE_GW | 0.016 | 1.804 | 0.059 | 2.771 |
| PAW_PBE | 0.015 | 1.789 | 0.060 | 2.813 |
| PAW_GGA | 0.013 | 1.794 | 0.059 | 2.658 |
| LDA_PP | 0.006 | 1.824 | 0.059 | 2.664 |
| USPP_GGA | 0.007 | 1.867 | 0.058 | 2.181 |
| USPP_LDA | 0.014 | 1.990 | 0.058 | 2.543 |

### 2.3. AlPO$_4$-34 Framework

#### 2.3.1. Energy-Volume Curve

The dependence of the total energy of the AlPO$_4$-34 framework on the unit cell volume is given in Figure 4. Seven E-V curves are all parabola, but the shape is a bit different. Four E-V curves from PBE_mGGA, PAW_PBE, PAW_GGA, and USPP-GGA are almost the same shape parabola, where the total energy at a volume of 750 Å$^3$ is larger than that at 900 Å$^3$. This trend is reversed for USPP_LDA, LDA_PP, and PBE_GW. The E-V curve of PBE_GW heavily deviates from PAW_PBE.

#### 2.3.2. Fitted Equilibrium Volume ($V_{0,fitted}$)

The rough equilibrium volume (Table 3) of AlPO$_4$-34 is obtained by fitting the E-V curve with the third order Birch–Murnaghan equation of state. PBE_mGGA gives rise to 826.90 Å$^3$, which is the closest to the experimental lattice volume on the dehydrated [63] HSAPO34 sample. Compared with the dehydrated [63]/hydrated [64] HSAPO34 sample, the calculated lattice volume of the AlPO$_4$-34 framework is $12.81 – 19.88 / 15.01 – 22.08$ Å$^3$ overestimated by PBE_GW, PAW_PBE, PAW_GGA, and USPP_GGA, while it is $11.35 / 9.15$ Å$^3$ roughly underestimated by LDA_PP.

**Table 3.** The lattice parameters of AlPO4-34 framework.

| Pseudopotential | a/Å | α/° | $V_0$/Å$^3$ | $V_{0,fitted}$/Å$^3$ |
|---|---|---|---|---|
| PBE_mGGA | 9.41 | 94.55 | 825.11 | 826.90 |
| PBE_GW | 9.47 | 94.35 | 840.84 | 837.27 |
| PAW_PBE | 9.46 | 94.40 | 839.85 | 840.78 |
| PAW_GGA | 9.47 | 94.40 | 841.39 | 842.27 |
| LDA_PP | 9.36 | 94.40 | 812.03 | 811.05 |
| USPP_GGA | 9.45 | 94.36 | 835.35 | 835.20 |
| USPP_LDA | 9.33 | 94.29 | 806.08 | 800.71 |
| Exp [63] | 9.40 | 94.27 | 822.39 | - |

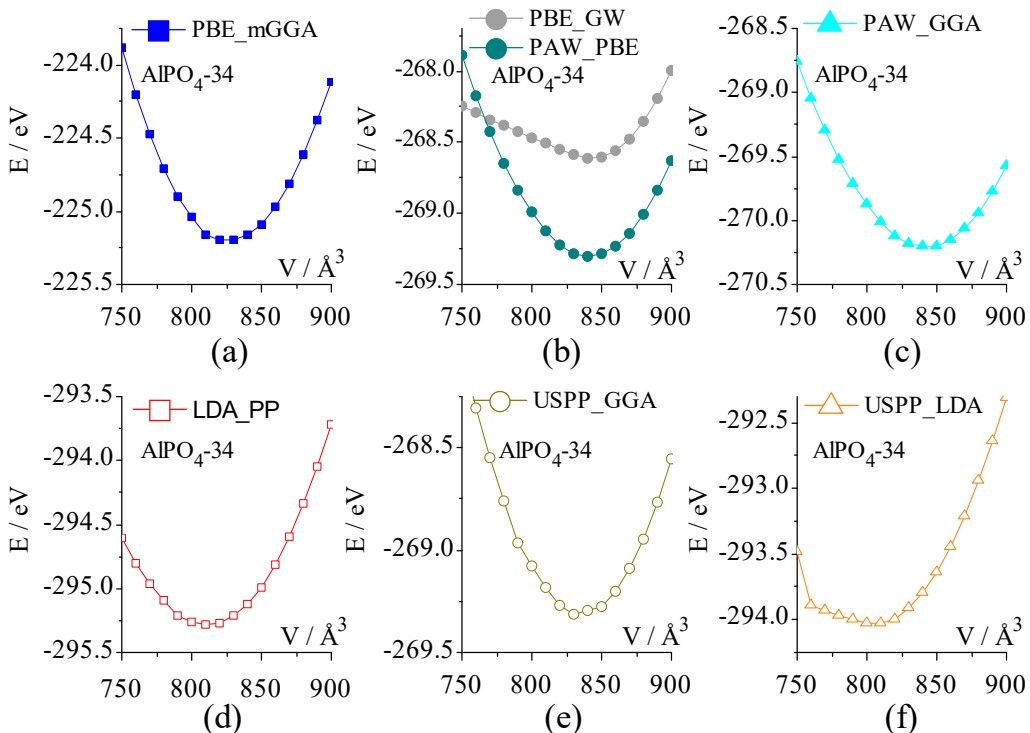

**Figure 4.** The total energy dependence on the unit cell volume of AlPO$_4$-34 was calculated by seven sets of pseudopotentials: (**a**) PBE_mGGA, (**b**) PBE_GW and PAW_PBE, (**c**) PAW_GGA, (**d**) LDA_PP, (**e**) USPP_GGA, and (**f**) USPP_LDA.

### 2.3.3. Precise Structure of AlPO$_4$-34 and Calculation Error

Table 3 also contains the more precise lattice parameters of the AlPO$_4$-34 framework, which are from the full optimization of the starting structure with the lattice volume near to $V_{0,\text{fitted}}$. During the relaxation, the force and the stress tensor are calculated, and the ions, the cell shape, and the cell volume are allowed to change.

To evaluate the calculation error, the neutron scattering experiment on the dehydrated HSAPO34 sample [63] is selected as the reference. As calculated, PBE_mGGA gives rise to the lattice volume, and the lattice edge is closest to the dehydrated HSAPO34 sample [63], followed by LDA_PP. The optimized structure of the AlPO$_4$-34 framework by PBE_mGGA is provided in the Supplementary Material. By calculation, the deviation of lattice-volume/lattice-edge is only 2.72 Å$^3$/0.01 Å from PBE_mGGA and $-10.36$ Å$^3$/$-0.04$ Å from LDA_PP, respectively. As to the deviation of the lattice angle, it is 0.28° from PBE_mGGA and 0.13° from LDA_PP, slightly higher than the lowest deviation. For the rest five sets of pseudopotentials, (1) the lattice volume deviation is big, ranging from 12.96–19.00 Å$^3$, assumedly increasing in order of USPP_GGA, USPP_LDA, PAW_PBE, PBE_GW, and PAW_GGA; (2) the absolute values of lattice edge deviations are assumed to be close to each other, ranging from 0.05–0.07 Å; and (3) lattice angle deviation ranges from 0.02° to 0.13°, assumedly changing in order of USPP_LDA < PBE_GW < USPP_GGA < PAW_GGA = PAW_PBE.

A previous powder neutron diffraction study showed that for the dehydrated HSAPO34 sample, the length was 1.651–1.812 Å for the Al-O bond and 1.500–1.589 Å for the P-O bond, and the angle was 143.564°–150.527° for ∠P-O-Al and 107.279°–110.795° for ∠O-T-O (see Table S2 in the Supplementary Material).

The bond length and the bond angle are two basic parameters reflecting the ion position and the bonding of crystalline. Figure 5 shows the absolute deviation of bond length and bond angle of the AlPO$_4$-34 framework. The deviation is defined as the theoretical value from each set of pseudopotentials minus the experimental data on the dehydrated HSAPO34 [63]. The dots around $X$-axis would vary with different sets of pseudopoten-

tials in a similar manner, but the jump magnitude is not different. As seen, those seven sets of pseudopotentials are capable of predicting either bond length or bond angle. No matter which pseudopotential is used, it is assumed that (1) the P-O(1) bond length is overestimated while the rest of the three P-O bonds are underestimated; (2) all Al-O bond lengths are overestimated, except the Al(1)-O(1) bond; (3) the absolute deviation of bond length is below 0.1 Å for either P-O bond or Al-O bond; (4) three angles (blue color) are with the weak absolute deviation: ∠O(1)-T-O(2), below 0.471°; ∠O(1)-T-O(3), below 0.723°; ∠O(1)-T-O(4), 1.120°–1.443°; (5) three angles (magenta color) are with the relatively larger absolute value of derivation: ∠T-O(1)-Al(1), 3.692°–4.862°; ∠T-O(4)-Al(4), 2.020°–6.306°; ∠O(2)-T-O(3): 3.217°–3.574°; and (6) the rest angles are with the absolute value of deviation below 3.00°.

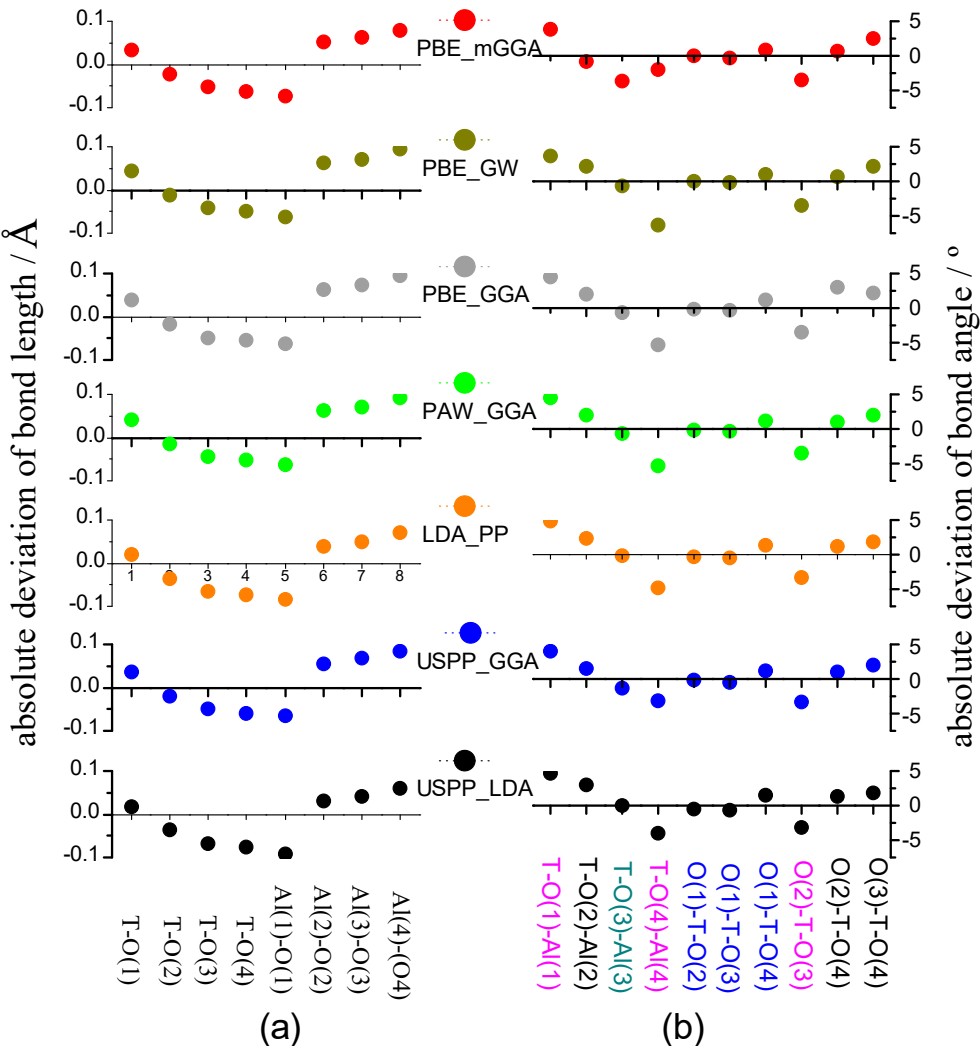

**Figure 5.** The calculation deviation of (**a**) bond length and (**b**) bond angle for the AlPO$_4$-34 framework (T = P).

The RMS deviation is further calculated to estimate the calculation error. As shown in Table 3, the RMS deviation ranges from 0.058 Å–0.066 Å for the bond length and 2.181°–2.813° for the bond angle. Among seven sets of pseudopotentials, PBE_mGGA gives rise to the second of the least RMS deviation of bond angle (i.e., 2.310°) and a small deviation of bond length (i.e., 0.066 Å). The rest six sets of pseudopotentials have almost the same RMS deviation of bond length. The RMS deviations of bond angles are a bit different from each other, for instance, 2.813° for PAW_PBE, 2.664° for LDA_PP, and so on.

In sum, results are reproducible with different levels of precision [25], in good agreement with the experiment [62,63]. Among seven DFAs, the comprehensive analysis indicates that PBE_mGGA probably provides the crystal structure of all-silica chabazite or AlPO$_4$-34 framework (Table S3 or Table S4) with the highest accuracy, followed by LDA_PP.

### 2.4. Calculation Time and Phonon Calculation

The calculation duration is important for the researcher to perform the PDFT calculation. Table 4 lists CPU times for the structural characterization of two model zeolites. The CPU time includes (1) the E-V curve scanning, (2) following full optimization, and (3) the phonon calculation.

**Table 4.** The calculation duration for the structural characterization of all-silica chabazite and AlPO$_4$-34 framework.

| Pseudopotentials | CPU Times (Seconds) | | | | | |
| | All-Silica Chabazite | | | AlPO$_4$-34 | | |
| | E-V Scan | Full-Opt * | Phonon | E-V Scan | Full-Opt * | Phonon |
| --- | --- | --- | --- | --- | --- | --- |
| PBE_mGGA | 31,803 | 553 | 25,980 | 91,328 | 718 | 24,890 |
| PBE_GW | 72,179 | 1449 | 23,842 | 178,857 | 1491 | 19,849 |
| PAW_PBE | 18,862 | 80 | 10,076 | 43,155 | 277 | 10,689 |
| PAW_GGA | 47,542 | 80 | 11,703 | 66,963 | 356 | 11,064 |
| LDA_PP | 15,824 | 310 | 9902 | 37,538 | 237 | 10,133 |
| USPP_GGA | 177,071 | 221 | 11,422 | 401,072 | 14,747 | 10,757 |
| USPP_LDA | 14,390 | 170 | 8438 | 31,803 | 285 | 9965 |

\* Starting geometries in the full optimization are the scan structure with the lattice volume near to the fitted value in Tables 1 and 3.

For the E-V curve scanning, taking AlPO$_4$-34 as an example, PBE_mGGA costs 91,328 s, which is 51.06% of the CPU times of PBE_GW and 22.77% of CPU times of USPP_GGA, respectively. The calculation duration obviously varies with the pseudopotential category. For all-silica chabazite, CPU times reduce in order of USPP_GGA > PAW_GW > PAW_GGA > PBE_mGGA > PAW_PBE > LDA_PP > USPP_LDA. In the case of AlPO$_4$-34, the calculation duration change trend is similar to all-silica chabazite, but the order of PBE_mGGA and PAW_GGA needs to interchange.

Compared with the respective E-V curve scanning, the full optimization needs rather less CPU times at each set of pseudopotential. The longest times are only about 1449 s for all-silica chabazite and 14,747 s for AlPO$_4$-34 framework. The main reason is caused by the reasonable selection of the starting geometry, which is with the lattice volume near to the fitted value (Tables 1 and 3). As known, the computation duration for completing geometry optimization depends largely on the quality of the initial structure.

Whether the E-V curve scanning and following full optimization are considered together, the CPU times order is the same as that for completing the E-V curve scanning. The CPU time difference is a bit big between the maximum and the minimum. Taking the AlPO$_4$-34 framework as an example, the longest task requires 401,072 s while the shortest task only needs 31,803 s. The former is 12.61 times as long as the latter. Therefore, it is meaningful to investigate the influence of the pseudopotential category on the computation duration.

Performing the phonon calculation is to further judge whether the optimized structure is in the equilibrium state or not. As shown in Table 4, for either all-silica chabazite or AlPO$_4$-34 framework, the CPU times for completing phonon calculation reduce in order of PBE_mGGA > PBE_GW > PAW_GGA > USPP_GGA > PAW_PBE > LDA_PP > USPP_LDA. The computation time from PBE_mGGA is also acceptable. For example, it would cost 24,890 s to complete the phonon calculation of AlPO$_4$-34 framework.

When the E-V curve scanning, the following full optimization, and the phonon calculation are considered together, it seems more meaningful to understand the influence of DFA category on the calculation duration. The analysis of all-silica chabazite indicates that the calculation duration reduces in order of USPP_GGA, PBE_GW, PAW_GGA, PBE_mGGA, PAW_PBE, LDA_PP, and USPP_LDA. In the case of AlPO$_4$-34 framework, the order is similar, but the order of PAW_GGA and PBE_mGGA needs to interchange. Whether the total CPU times from USPP_LDA is regarded as 1.0, taking AlPO$_4$-34 framework as an example, the total CPU times from other DFAs are following, USPP_GGA: 10.1, PBE_GW: 4.8, PBE_mGGA: 2.8, PAW_GGA: 1.9, PAW_PBE: 1.3, LDA_PP: 1.1. It is worth noting that, for either all-silica or AlPO$_4$-34 framework, USPP_LDA costs the least calculation time; the calculation times from LDA_PP or PBE_mGGA are also acceptable since they are only 1.1 times or 2.5-2.8 times of the least.

The computer used is the 2-socket tower server of Dell EMC PowerEdge T640. The PowerEdge T640 features the second Generation Intel (R) xeon® Processor scalable family. The processor name is the Intel (R) xeon® Silver 4110, and the memory capacity is 64 Gb. All calculations are computed in the single node. It is worthnoting that the computation durations are directly determined by the computer configuration and the computer architecture.

## 3. Materials and Methods

All calculations are performed by the Vienna Ab initio Simulation Packages (VASP) [65]. Seven sets of typical pseudopotentials supplied with VASP packages are used to calculate porous materials. They are (a) strongly constrained and appropriately normed (SCAN) meta-generalized-gradient approximation (meta-GGA) [66,67] with the implemented Perdew–Burke–Ernzerh [68] of functional (PBE-mGGA), (b) the GW potentials [69] with the implemented Perdew–Burke–Ernzerh of functional (PBE_GW), (c) the projector augmented wave type with the implemented Perdew–Burke–Ernzerh [68] of functional (PAW_PBE), (d) the projector augmented wave (PAW) [37,70] type with the generalized gradient approximation (PAW_GGA), (e) the standard pseudopotential with the local density approximation of (LDA_PP), (f) the ultrasoft type with the generalized gradient approximation (USPP_GGA) [71,72], and (g) the ultrasoft type with the local density approximation (USPP_LDA). For detailed information, refer to Table S5 in the Supplementary Material. The variables for "TITLE" and "TEXCH" in the pseudopotential file (i.e., POTCAR) are used to differentiate the version.

The all-silica chabazite structure is initially obtained from Internal Zeolite Association (IZA) [73]. The AlPO$_4$-34 zeolite structure is constructed by the isomorphous substitution of Si$^{4+}$ in all-silica CHA framework by Al$^{3+}$ or P$^{5+}$ ion via the SMIII mechanism [74], making Al$^{3+}$ and P$^{5+}$ ions appear alternatively [75]. All calculations are under periodic boundary conditions, and the energy cutoff is set to 900 eV. The convergence threshold selected is $1 \times 10^{-2}$ eV/Å for the atomic force and $1 \times 10^{-7}$ eV for the energy. The Brillouin-zone sampling is restricted to $1 \times 1 \times 1$.

The equilibrium structure is calculated by each set of pseudopotentials through two steps, avoiding the possible problem of the incompleteness of the plane wave basis. For either all-silica chabazite or AlPO$_4$-34 framework, (1) a series of structures are partially optimized at the constant volume by relaxing the cell shape and the ion position. The constant volume is selected from 750 to 900 Å$^3$ with an interval of 10 Å$^3$. The total electron energy and the lattice volume are abstract from the output file of partial optimization, which is used to draw the energy-volume (E-V) curve. Fitting the E-V curve with the Birch–Murnaghan equation of state gives rise to the rough value of lattice volume (V$_{0,\text{fitted}}$) and total energy (E$_0$) in the equilibrium state. (2) The full optimization calculation is carried out by relaxing the cell shape, the ionic coordinates, and the volume. The atomic force and the stress tensor are also calculated. The initial structure for the full optimization is with a lattice volume near to a fitted value from the E-V curve. The optimized structure is used to analyze the vibration frequency during the phonon calculation.

## 4. Conclusions

To evaluate the prediction feasibility of the crystal structure of unknown microporous materials, the CHA framework of chabazite and HSAPO34 have been structurally simulated by PBE_mGGA, PBE_GW, PAW_PBE, PAW_GGA, LDA_PP, USPP_GGA, and USPP_LDA. The calculation results on the testing framework allow us to draw the following conclusions. (1) Each density functional approximation is capable of providing the lattice volume, the lattice edge, the bond length, and the bond angle, with the different deviations from the neutron scattering experiment data. (2) Lattice volume is about 16.69–19.00 $Å^3$ overestimated by PBE_GW, PAW_PBE, and PAW_GGA, while it is about 16.31–17.00 $Å^3$ underestimated by USPP_LDA. The least deviationof lattice volume is below 5.18 $Å^3$ from PBE_mGGA, followed by LDA_PP. (3) RMS deviation from each DFA approach is less than 0.016 Å for bond length and less than 2.813° for bond angle. (4) The CPU times for completing the crystal structure calculation reduce in order of USPP_GGA > PBE_GW > PAW_GGA, PBE_mGGA > PAW_PBE > LDA_PP > USPP_LDA. (5) Whether the researcher does not mind that the calculation time is long or not, among those seven approximations, PBE_mGGA is the best choice to predict the lattice structure with the highest calculation precision for either all-silica chabazite or $AlPO_4$-34 framework. (6) The tradeoff between the calculation precision and the computation efficiency indicates that LDA_PP is a preferred approximation method capable of providing the lattice structure of either all-silica chabazite or $AlPO_4$-34 framework with a higher calculation precision and a lower computation cost. (7) For unknown microporous materials, one might first construct the rough topology through the structure combination of the tetrahedral units of $SiO_4$ or/and $AlO_4$, or/and $PO_4$, or composite building units; second, they would search for the possible equilibrium crystal structure through E-V scanning and energy minima; ultimately, they would identify the crystal structure as the reasonable energy minima by the phonon calculation.

**Supplementary Materials:** The following supporting information can be downloaded at: https://www.mdpi.com/article/10.3390/inorganics11050215/s1. Table S1 Geometric parameters of zeolite chabazite HSSZ-13 (T = Si) reported in the neutron scattering experiment. Table S2 Geometrical parameters of Zeolite HSAPO34 (T = P) reported in the neutron scattering experiment. Table S3 The structure of all-silica chabazite optimized by PBE_mGGA. Table S4 The structure of $AlPO_4$-34 framework optimized by PBE_mGGA. Table S5 The information on seven sets of pseudopotentials selected in this study.

**Funding:** This research was funded by Shandong University (the Basic Research Program (No. 2019GN018)).

**Data Availability Statement:** Please refer to Supplementary Materials.

**Conflicts of Interest:** The authors declare no conflict of interest.

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
