# Peer review of "Periodic Density Functional Theory (PDFT) Simulating Crystal Structures with Microporous CHA Framework: An Accuracy and Efficiency Study"

_inorganics, doi:10.3390/inorganics11050215_

Round 1

Reviewer 1 Report

In this work Chen evaluates different periodic DFT levels of theory for the modelisation of the microporous structures of chabazite and aluminium phosphate.

This research could be useful for theoretical studies concerning the adsorption of different pollutants by zeolite and AlPO4. For this reason we consider that it deserves to be published in Inorganics with minor revisions.

In the conclusion the author should clearly state for each framework, what is the best option that offer accuracy at a reasonable computational cost.

The choice of the seven density functional approximations should be justified. Some references about of their performance toward other structures could be of interest.

There are several typos along the article:

"... the artificial microporous materials have received the fast development"

"... powder sample could provide the useful information such as atomic coorinates ..."

"worthnoting "

"... when the theoretical value at each set of pseudopotential minus the corresponding experiment value."

"To evoluating the prediction feasibility ..."

"Duration of calculation" should be replaced by "Calculation time" along the article.

Relevant studies about the modelisation of zeolites by DFT should be mentioned:

Nanomaterials 2019, 9(5), 715

Phys. Chem. Chem. Phys., 2022,24, 24992-24998

It should be improved.

Author Response

Thank you for your spending the time helping me. All of your suggestions are helpful. 

Following your suggestions, I have made the major modifications as follows.

1) I have already pointed out which method is preferred.

2) I have justified the choice of approximation methods.

3) I have added some references (Refs 22-24, Refs 1d and 18b). 

4) I have corrected several typos and grammatic errors.

Please see the file called as "Response-to-Reviewer-1.pdf"

Reviewer 2 Report

The work of a Chinese scientist describes the use of a quantum chemical modeling approach to be able to predict useful applied properties of microporous crystals with SNA. The author quite well describes the relevance of his work and the rationale for the selection of research objects, namely zeolites of chabazite HSSZ-13 and HSAPO34. There are also no questions about the methodology of the experiment. The results obtained in this work are of fundamental importance. It is also worth noting the original way of writing the manuscript, which attracts the attention of the reader. Before the publication of the work there are only small remarks. Thus, Equation 1 must be given in a higher quality. Also, unfortunately, for some reason in the work (in the submission) there is no file with the data of "Supplementary Materials" which greatly complicates the final interpretation of the work. The reviewer has no doubts about the reliability of the results, but this file must be attached to the work.

Author Response

Thank you for your spending the time helping me. Your suggestions are helpful. 

Following your suggestions, I have made the corresponding modifications as follows.

1) I have rewritten the Equation 1, and made it have a higher quality.

2) The file of “Supplementary Materials” has been uploaded already.

Reviewer 3 Report

The manuscript "Periodic Density Functional Theory (PDFT) Simulating Crystal Structures with Microporous CHA Framework: an Accuracy and Efficiency Study" from Xiao-Fang Chen used several density functional approximations (DFAs) to simulate the crystal structure of microporous material and compared the results with the existing experimental results. In my opinion, the manuscript should be improved before being published. The author should clearly state why he choses those seven methods, what their limitations are, and which method should be preferred. 

Also, the manuscript should be revised to correct English errors.

Author Response

Answer: Thank you for spending your precious time reviewing my manuscript. Thanks for your constructive suggestions.

Following your suggestions, I have added the explanation of the method selection, and the respective method limitation in the section of “1. Introduction”.

As for the preferred method, I stated it in the section of “Abstract” and “4. Conclusion”.

Round 2

Reviewer 3 Report

The authors have answered the questions, the manuscript may now be published.